# Sensitivity of Immunodiagnostic Tests in Localized Versus Disseminated Tuberculosis—A Systematic Review of Individual Patient Data

**DOI:** 10.3390/tropicalmed10030070

**Published:** 2025-03-07

**Authors:** Michael Eisenhut, Shagun Shah, Ozge Kaba, Manolya Kara, Murat Sütçü, Kyoung-Ho Song, Hong Bin Kim, Maoshui Wang

**Affiliations:** 1Paediatric Department, Luton & Dunstable University Hospital, Lewsey Road, Luton LU40DZ, UK; 2Pediatric Infectious Diseases Department, Başakşehir Çam and Sakura City Hospital, 34480 Istanbul, Turkey; ozgekabamd@gmail.com; 3Department of Pediatric Infectious Diseases, School of Medicine, Yeditepe University, 34728 Istanbul, Turkey; 4Department of Pediatric Infectious Diseases, School of Medicine, İstinye University, Bahçeşehir Liv Training and Education Hospital, 34010 Istanbul, Turkey; 5Division of Infectious Diseases, Department of Internal Medicine, Seoul National University Bundang Hospital, Seoul National University College of Medicine, Seongnam 13620, Republic of Korea; 6Shandong Key Laboratory of Infectious Respiratory Disease, Jinan 250013, China

**Keywords:** miliary tuberculosis, tuberculous meningitis, lymph node tuberculosis, interferon gamma release assay, sensitivity

## Abstract

Our objective was to perform a systematic review of individual patient data comparing immunodiagnostic test sensitivity in patients with localized versus disseminated tuberculosis who are from high- and less-than-high-income countries. In a systematic review of individual patient data, we compared IGRA results and characteristics of patients with disseminated tuberculosis with IGRA results and characteristics of patients with localized tuberculosis. Data were extracted from Pubmed, EMBASE and the Cochrane Library, analyzed and presented following the PRISMA-IPD and STROBE statements. We identified 52 patients with localized and 105 with disseminated tuberculosis. Immunodiagnostic tests in localized tuberculosis from high-income countries were positive in 88.8% and in 67.3% of patients with disseminated tuberculosis (*p* = 0.034). In patients from less-than-high-income countries, the sensitivity of immunodiagnostic tests was not significantly lower with disseminated tuberculosis. Patients with disseminated tuberculosis were significantly younger and had a higher rate of microbiological confirmation. Multivariate logistic regression analysis revealed that rate of microbiological confirmation was associated with a negative IGRA. Disseminated tuberculosis may be associated with a reduced sensitivity of IGRA in high-income countries and this may be related to a higher bacterial load with a negative IGRA.

## 1. Introduction

Immunodiagnostic testing may be a useful adjunct in the diagnosis of active tuberculosis in patients where the microbiological confirmation has a low rate of positivity, particularly in children with miliary tuberculosis or lymph node tuberculosis [1,2].

Previous studies contained data on the sensitivity of tuberculin skin tests or interferon gamma release assays (IGRAs) for *Mycobacterium tuberculosis* (*M. tuberculosis*) infection in different forms of tuberculosis. None of the previous studies compared the sensitivity of immunodiagnostic tests in disseminated versus localized forms of tuberculosis.


**Rationale:**


Previous systematic reviews of IGRA results in patients with lymph node tuberculosis showed them to yield a positive result in between 89 and 95% [3,4]. In miliary tuberculosis, systematic reviews targeting this condition are lacking but retrospectively analyzed data sets like the 44 patients with miliary tuberculosis investigated in South Korea [5] and the 549 patients with miliary tuberculosis from the US tuberculosis cases reported to the Centers for Disease Control and Prevention in the USA [6] showed positive IGRA results in between 68 and 75% of cases.

The variability of test sensitivity of immunodiagnostic testing in active tuberculosis may be due to nutritional factors which have previously been shown to reduce tuberculin skin test and IGRA sensitivity in patients with active tuberculosis and a body mass index of less than 16 [7] and may account for the lower sensitivity of tuberculin skin tests and IGRA’s in low-income countries [8].

Other factors potentially influencing reactivity in immunodiagnostic testing include immunocompromising disease and medication suppressing the immune response. Laboratory parameters found to be associated with negative IGRA results in disseminated tuberculosis include a lower lymphocyte count, lower hemoglobin levels and higher C-reactive protein levels [5,9].


**Objectives:**
To perform a systematic review of individual patient data comparing immunodiagnostic test sensitivity in patients with localized versus disseminated tuberculosis who are from high-income countries.To perform a systematic review of individual patient data comparing immunodiagnostic test sensitivity in patients with localized versus disseminated tuberculosis who are from less-than-high-income countries.To identify risk factors for false-negative immunodiagnostic tests in patients with active tuberculosis.


## 2. Materials and Methods

The systematic review and meta-analysis of individual participant data was conducted following the Preferred Reporting Items for a systematic review and meta-analysis of individual participant data (PRISMA-IPD) and the STROBE Statement for case–control studies [10]. This approach, which is tailored to systematic reviews of trials, was adapted to this systematic review of data on diagnostic sensitivity.


**Protocol and registration:**


The protocol for this systematic review and meta-analysis of individual participant data was pre-registered on the PROSPERO International prospective register of systematic reviews of the National Institute of Health and Care Research of the United Kingdom, registration number CRD42021292332, under the title: “Sensitivity of immunodiagnostic test in lymph node versus miliary and central nervous system tuberculosis-an analysis of individual patient data”.


**Eligibility criteria:**


Eligibility criteria for inclusion in this systematic review included all patients with either isolated lymph node tuberculosis or miliary and/or central nervous system tuberculosis, who had a valid IGRA result for *M. tuberculosis* infection (positive or negative) or if an indeterminate IGRA result that was obtained had an unequivocal tuberculin skin test result (equal to or greater than 5 mm being positive regardless of previous BCG immunization, and negative if less than 5 mm in diameter).


**Rationale for criteria:**


The criteria were chosen as a localized tuberculosis is best defined in lymph node tuberculosis with a higher probability of histological confirmation. In addition, considering the generally accepted fact that miliary tuberculosis is disseminated, central nervous system tuberculosis was also regarded as disseminated because of its association with miliary tuberculosis even with a normal CT scan of the chest [11].


**Inclusion criteria:**

**Case definitions:**


Definition of isolated lymph node tuberculosis:

The definition of lymph node tuberculosis was considered an extension of the definition by Cantrell et al. of cervical tuberculous lymphadenitis [12]:

Three or more of the following six criteria had to be satisfied: (1) an enlarged lymph node, (2) a positive reaction for a tuberculin skin test or IGRA, (3) a compatible histopathological appearance defined as epithelioid cell granuloma with central caseous necrosis, (4) a demonstration of acid fast bacilli (AFB) in the biopsy specimen, (5) the growth of *M. tuberculosis* from the biopsy specimen or a positive polymerase chain reaction for *M. tuberculosis* from tissue, and (6) a definite response to anti-tuberculosis chemotherapy.

Definition of miliary tuberculosis:

We have used the definition from Sharma and Mohan: “Diagnosis of miliary TB requires the presence of a diffuse miliary infiltrate on a chest radiograph or high-resolution CT (HRCT) or histopathological evidence of miliary tubercles in tissue specimens obtained from multiple organs”. The term “miliary”, as the authors explained, is used “to describe the resemblance of gross pathological findings of innumerable millet seeds in size and appearance”, where a “pattern on a chest radiograph” has been defined as “a collection of tiny discrete pulmonary opacities that are generally uniform in size and widespread in distribution, each of which measures two mm or less in diameter”, and the authors add that “in 10% of the cases, the nodules may be greater than 3 mm in diameter” [13].

Definition of tuberculous meningitis:

We included both patients with definite and highly probable TBM. According to predefined criteria used in previous studies [14,15], patients were diagnosed as definitely having TBM if blood culture, CSF culture or PCR results were positive for *Mycobacterium tuberculosis*, or as highly probably having TBM if they had clinical features of chronic meningitis which included fever, headache and neck stiffness for more than 2 weeks and at least 3 supporting criteria consisting of (1) CSF findings of lymphocytic pleocytosis, raised protein levels and sterile cultures, (2) CT/MRI findings of hydrocephalus, granulomas or basal exudates, (3) evidence of extraneural tuberculosis, and an (4) appropriate response to anti-tuberculosis chemotherapy.


**Exclusion criteria:**
There was no clear immunodiagnostic result.The immunodiagnostic result was obtained prior to symptom onset and prior to diagnosis.The illness did not conform to the case definitions.



**Identifying studies, information sources and search strategy:**


The databases of Pubmed, EMBASE and the Cochrane library from its inception were searched.

For all databases, the keyword combination using Boolean operators was (interferon gamma release assay OR IGRA OR T spot TB OR quantiferon) AND (lymph node tuberculosis OR tuberculous lymphadenitis OR miliary tuberculosis OR cerebral tuberculoma OR tuberculous meningitis). 


**Study selection process:**


The study selection process was conducted by Dr. Eisenhut and Dr. Shah independently and documented in the PRISMA flow diagram. Data from full case reports, case series and conference abstracts were included.


**Data collection processes:**


All five authors of this study were involved in data collection and the lead author did the screening and double checking of all data for eligibility. The researchers retrieved the data from the manuscripts identified and transferred them onto Excel files.

For studies where individual patient data were not documented in the manuscript, the authors were asked to provide the available data in an Excel spread sheet and e-mail them to the lead author for screening and inclusion if they fulfilled the inclusion criteria.

**Data items:** We extracted data on age, gender, country of origin, IGRA for *M. tuberculosis* infection and HIV test results at diagnosis, results of imaging, histopathology and results of nucleic acid amplification tests (NAATs) and the culture of any site obtained. Microbiological confirmation of tuberculosis was defined as a positive NAAT or culture result. Any laboratory parameters obtained were recorded. The response to treatment was recorded.


**Specification of outcomes and effect measures:**


The primary outcome was differences in IGRA results in patients with localized versus disseminated tuberculosis in high-income countries and in less-than-high-income countries.

The secondary outcomes investigated were the influence of age, gender, immunosuppressive co-morbidity, microbiological confirmation and HIV infection between patients with localized versus disseminated tuberculosis and, in addition, for the comparison of patients with positive versus negative immunodiagnostic testing, country of origin, hemoglobin level, lymphocyte count, erythrocyte sedimentation rate (ESR) and C-reactive protein levels.


**Statistical methods:**


Before the employment of statistical testing, data were explored regarding parametric or non-parametric distribution characteristics. If data were found to be significantly skewed or to have a standard deviation exceeding 60% of the arithmetic mean, indicating outliers, median and range were used for summary statistics of continuous data and the Mann–Witney U test was used.

SSPS version 27 (IBM, Armonk, NY, USA) was used for data exploration and the analysis of continuous data.

For categorical data, the chi-square test was used and it was used with Yates correction if a subgroup had less than 30 participants. Fisher’s exact test was used if one group had less than 5 participants. Epi-info version 7.0 (CDC, Atlanta, GA, USA) was used for the analysis of categorical data.

For personal characteristics showing a significant difference between patients with and without a positive IGRA, a multivariate logistic regression analysis (using SSPS version 27 (IBM)) was conducted, entering items that were significantly different on univariate analysis between groups as independent variables in the model and IGRA status as the dependent variable.

A *p*-value of <0.05 was used to indicate a statistically significant difference. The 2-sided *p*-value was used with the exception of the comparison of IGRA results between disseminated and localized tuberculosis as, following the literature review and sample size calculation, a higher IGRA positivity rate for localized tuberculosis was anticipated. For this comparison, a one-sided *p*-value was chosen.


**Exploration of variation in effects:**


We assessed the variation in immunodiagnostic test results introduced by country of origin using subgroup analyses, analyzing the results separately for patients from high- compared to less-than-high-income countries following the World Bank definition [16].

We assessed the variation in immunodiagnostic test results in children (<16 years of age) and adults separately.

We assessed the variation in immunodiagnostic test results for microbiologically confirmed tuberculosis in disseminated versus localized tuberculosis in high- and less-than-high-income countries separately.

We assessed the differences in the characteristics of patients with disseminated versus localized tuberculosis by using univariate followed by multivariate analyses comparing patients with positive and negative immunodiagnostic tests.


**Risk of bias across studies:**


We discussed the risks of publication bias that were inherent in the data extracted for this study, which used exclusively case reports and case series. Positive-results bias, a type of publication bias, occurs when authors are more likely to submit, or editors are more likely to accept, positive results than negative or inconclusive results. Across all studies, it is likely that case reports with a definitive diagnosis were over-represented as more likely to be accepted for publication.

Data bias may be indicated by the fact that there was a difference in the completeness of data reported between the groups analyzed and compared. Incompleteness of data was documented for all data where present and analyzed statistically for items where there were statistically significant differences between groups and the results of the analysis was discussed as a source of data bias.


**Sample size calculation:**


We performed a sample size calculation to determine how many patients were required in the groups of disseminated tuberculosis and localized lymph node tuberculosis using data from high-income countries (South Korea and Taiwan). One study on IGRA results in patients with miliary tuberculosis with a high rate of microbiological confirmation (95%) showed a sensitivity of 68% [5]. For lymph node tuberculosis, the sensitivity of IGRA in another study was found to be 95% [4].

The calculation was based on the formula for binary outcome superiority trials with the assumption of a higher sensitivity in localized versus disseminated tuberculosis:n = f(α/2, β) × [p_1_ × (100 − p_1_) + p_2_ × (100 − p_2_)]/(p_2_ − p_1_)^2^
where p_1_ and p_2_ are the percent positive IGRA in the miliary tuberculosis and lymph node tuberculosis group, respectively, and f(α/2, β) is calculated from the type I error probability (α) and the type II error probability β taking into account the standard deviation of the percentage and the smallest difference in percentages that we regarded as important to detect.

For the output, the power to detect a statistically significant difference was set at 80% (β = 10%) at a significance level of α = 5% [17].

The sample size calculation yielded the following result:

For equally sized groups, at least 29 patients in each group are required (total of n = 58) to have a 80% chance of detecting, at the 5% significance level, a difference of at least 27% in sensitivity of IGRA between groups. For the expected size difference in groups, with the group of miliary tuberculosis being larger, using a 2:1 ratio, the smaller group was calculated to have at least n = 21 patients and the sample size of the larger group was calculated to have at least n = 44 [18].

## 3. Results

### 3.1. Study Selection and IPD Obtainment

The results of study selection are documented in a PRISMA flow diagram (Figure 1). We documented compliance with the PRISMA-IPD and STROBE statements in the checklists in Appendix A. We obtained individual patient data from a total of 157 patients with tuberculosis. A total of 52 patients had a diagnosis of lymph node tuberculosis [19,20,21,22,23,24,25,26,27,28,29,30,31,32,33,34,35,36,37,38,39,40,41,42,43,44], and 105 had a diagnosis of miliary tuberculosis and/or tuberculous meningitis [13,45,46,47,48,49,50,51,52,53,54,55,56,57,58,59,60,61,62,63,64,65,66,67,68,69,70,71,72,73,74,75,76,77,78,79,80,81,82,83,84,85,86,87,88,89,90,91,92,93,94,95,96,97,98,99,100,101,102,103,104,105,106,107,108,109,110,111,112,113,114,115,116,117,118,119]. We were able to gather additional individual patient data from large studies by collaboration with authors from South Korea, China and Turkey.

### 3.2. Study Characteristics

We included patients from the following high-income countries: For disseminated tuberculosis (n = 52), Japan had 11 patients included, USA had 8, Portugal had 6, Australia had 5, the United Kingdom had 5, Germany had 4, Spain had 3, Korea had 3, Italy had 2, Romania had 2, Poland had 1, Denmark had 1, and Greece had 1.

For lymph node tuberculosis (n = 27), Japan had 17 patients that were included, France had 3, Germany had 2, Poland had 1, Portugal had 1, Taiwan had 1, Italy had 1, and USA had 1.

We obtained data from the following numbers of patients from less-than-high-income countries: For disseminated tuberculosis (n = 43), we obtained data from 21 patients from China, 12 from Turkey, 6 from India, 1 from Eritrea, 1 from Indonesia, 2 from Guatemala, 1 from Guinea Bissau, 1 from Malaysia, 1 from Iran, 1 from Egypt, and 1 from Mexico. For lymph node tuberculosis (n= 24), we obtained data from 10 patients from Turkey, 3 from China, 3 from Morocco, 2 from Guinea, 1 from the Congo, 1 from Ethiopia, 1 from Somalia, 1 from Nepal, 1 from Gabon, and 1 from Peru.

### 3.3. Results of Syntheses

#### 3.3.1. High-Income Versus Less-Than-High-Income Countries

Personal characteristics were listed and compared between patients with localized and disseminated tuberculosis in Table 1 for those from high-income countries and Table 2 for those from less-than-high-income countries.

Patients from high-income countries who had disseminated tuberculosis were significantly younger, had a higher rate of microbiological confirmation and a lower rate of positive IGRA results, indicating a lower sensitivity of IGRA in patients with this background. Immunodiagnostic tests in patients with localized tuberculosis from high-income countries were positive in 88.8% (95% confidence interval in 77% to 100%) and in 67.3% (95% confidence interval in 54% to 80%) of patients with disseminated tuberculosis. In patients from less-than-high-income countries, there was no significant difference between groups for any characteristics obtained. For types of co-morbidity associated with immunosuppression, we found no difference in numbers between groups from high- or less-than-high-income countries as shown in Appendix A. Co-morbidity was significantly more common in patients from high-income countries (28%) versus those from less-than-high-income countries by 8.3% (*p* = 0.006).

Table 3 shows the analysis of the relationship of patient characteristics to IGRA status (positive versus negative).

In the multiple logistic regression analysis entering microbiological confirmation and age as independent variables and IGRA positivity as a dependent variable, microbiological confirmation remained significantly associated with a negative IGRA result with a beta of 0.217, a t of 2.63 and a *p*-value of 0.010.

#### 3.3.2. Adults Versus Children

To further explore the influence of age on IGRA sensitivity, we compared the sensitivity of immunodiagnostic testing in disseminated versus localized tuberculosis separately in adults and children. We found that in adults, the sensitivity of immunodiagnostic testing was 67.9% in disseminated and 92.1% in localized tuberculosis (*p* = 0.002). In children, the sensitivity was 67.2% in disseminated and 46.1% in localized tuberculosis (*p* = 0.271).

#### 3.3.3. Subgroup Analysis of Microbiologically Confirmed Tuberculosis

In patients from high-income countries with microbiologically confirmed tuberculosis, immunodiagnostic test sensitivity was 72% in disseminated and 88% in localized tuberculosis (*p* = 0.147). In patients from less-than-high-income countries with microbiologically confirmed tuberculosis, immunodiagnostic test sensitivity was 76% in disseminated and 73% in localized tuberculosis (*p* = 0.539).

### 3.4. Risk of Bias Across Studies

We investigated for evidence of data bias for the characteristics that were significantly different between groups. Age was different in groups, but there was no significant difference between the reporting of age in groups with positive IGRA 114/120 and negative IGRA with 37/37 (*p* = 0.336). Of patients, 5/6 with age not reported were from less-than-high-income countries. There was no difference in the completeness of reporting on microbiological confirmation between localized and disseminated tuberculosis in patients from high-income countries (*p* = 1.000) and between patients with positive and negative IGRAs (*p* = 0.131).

## 4. Discussion

In patients from high-income countries, patients with disseminated tuberculosis had a significantly lower rate of positive IGRA results. In patients from less-than-high-income countries, there was no significant difference in the outcome of immunodiagnostic tests comparing patients with localized with disseminated tuberculosis. In a subgroup analysis restricted to patients with microbiologically confirmed tuberculosis, the rate of positive IGRAs was lower in disseminated tuberculosis compared to localized tuberculosis in those from high-income countries but this did not reach statistical significance. The latter was likely due to the smaller sample size. Regarding the comparison of IGRA results in disseminated versus other forms of tuberculosis, a previous multicenter study using data from collaborators from mainly high-income European countries found that in children with microbiologically confirmed tuberculosis IGRA sensitivity, when results were pooled for different versions of IGRAs ,was 70% (56/79) in those with disseminated versus 89% (283/315) in those with pulmonary tuberculosis, which we calculated was a statistically highly significant difference (*p* = 0.00003). This study also showed no significant difference in the sensitivity of IGRAs between different ethnic groups (Caucasian, African, Arabic or Latin American descent) which may mean that the differences in results we found for high- versus less-than-high-income countries were not due to ethnic backgrounds but environmental conditions like nutritional status impacting the immune response [119]. We found that in adults there was, but in children, there was not a difference in sensitivity in immunodiagnostic testing between disseminated and localized tuberculosis. This may have been due to the fact that the majority of children (73%) were from less-than-high-income countries where there was no difference in the sensitivity of immunodiagnostic test results between disseminated and localized tuberculosis.

Our results showed no association of immunosuppressive conditions with negative IGRAs. This is in contrast with previous studies summarized in a systematic review, which showed more negative IGRA results in patients with immunosuppressive conditions [120]. We also found that a younger age was associated with a lower positive IGRA result rate on univariate analysis, while this systematic review found that an older age was associated with a negative IGRA result. In our review, patients with disseminated tuberculosis had a higher rate of microbiological confirmation than patients with localized tuberculosis. Only the rate of microbiological confirmation was independently associated with a negative IGRA. This indicated that a higher bacterial load, which facilitates microbiological confirmation, may be involved in a reduced positivity of IGRAs. This is in alignment with the results of a previous study in HIV-infected patients with tuberculosis who had increased IGRA responses with lower bacillary loads [121]. In vitro studies shed a light on the underlying mechanisms: *M. tuberculosis* induces transforming growth factor-1 (TGF-1) expression and this cytokine suppresses interferon gamma production [122]. *M. tuberculosis* also suppresses interferon gamma production by silencing cytokine gene expression through epigenetic mechanisms. This has been established for DNA methylation [123]. The DNA methylation status in the peripheral blood mononuclear cells and T-lymphocytes of patients with tuberculosis and their asymptomatic household contacts were compared. Patients with TB were found to have DNA hypermethylation of the IL-2/STAT5, TNF/NF-κB, and IFN-γ signaling pathways. Multiple genes of the IL-12/IFN-γ signaling pathway (IL12B, IL12RB2, TYK2, IFNGR1, JAK1, and JAK2) were hypermethylated and thereby silenced in patients with TB. Alternatively, the abnormality of the immune system which causes a higher bacterial load to develop also leads to a reduced interferon gamma release in lymphocytes in response to *M. tuberculosis* antigens. We did not find a difference in lymphocyte numbers between groups, but the study may have been underpowered to investigate an association of lymphocyte count with IGRA response.

### Strengths and Limitations

A strength of this study is that we were able to ascertain data in numbers that were sufficient for, according to the sample size calculation, an adequately powered comparison of IGRA results in patients from both high- and less-than-high-income countries, comparing disseminated with localized tuberculosis. We were able to gather additional individual patient data from large studies. This increased data completeness added power to the results of this study.

A weakness was that due to the retrospective nature of the study, we were only able to include laboratory data on hemoglobin, lymphocyte counts, ESR and C-reactive protein levels from a minority of patients and the lack of statistically significant difference between the groups may have been due to a lack of statistical power.

Very few patients had HIV tests documented. This is unlikely to be relevant for the finding of a significant difference in IGRA results between groups from high-income countries as the HIV prevalence in patients with tuberculosis in those countries is generally very low.

## 5. Conclusions

Disseminated tuberculosis may be associated with a reduced rate of a reactive IGRAs compared to localized tuberculosis in high-income countries but not in countries with less-than-high incomes and this may be related to a higher bacterial load.

## Figures and Tables

**Figure 1 tropicalmed-10-00070-f001:**
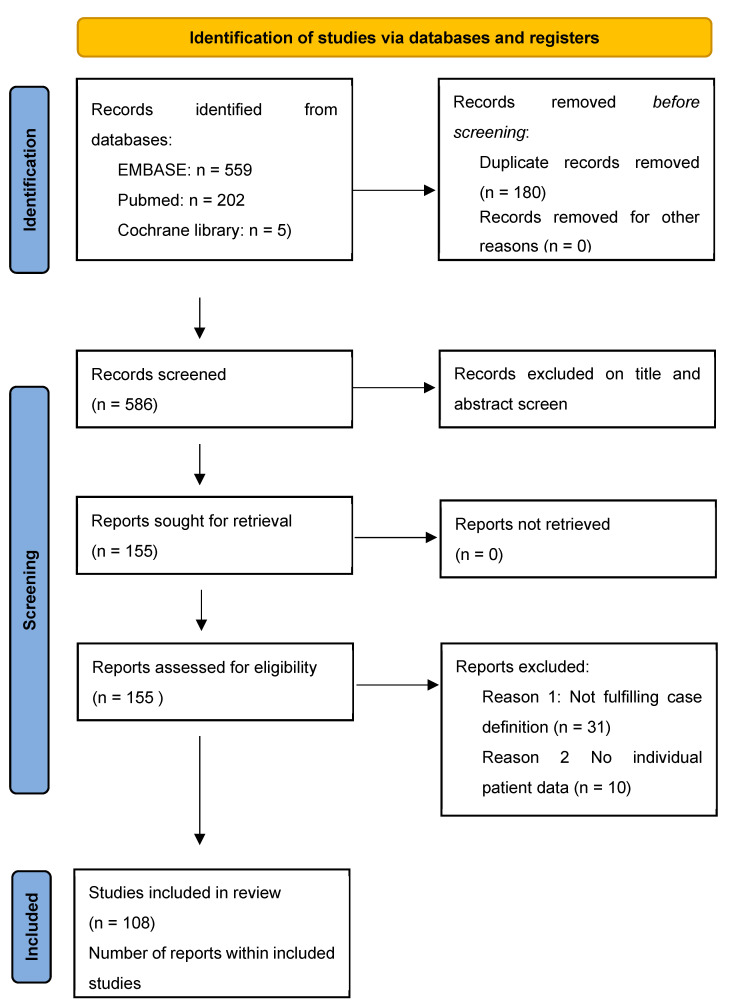
PRISMA flow diagram (August 2023).

**Table 1 tropicalmed-10-00070-t001:** Personal characteristics of patients with isolated lymph node tuberculosis and patients with miliary and/or central nervous system tuberculosis from high-income countries.

	Patients with Lymph Node Tuberculosis	Patients with Miliary and/or Central Nervous System Tuberculosis	*p*-Value for Comparison Between Groups
Age (years, median (range))	53.0 (4 to 89)	28.5 (0.04 to 87)	0.004
Gender (male/total number of patients for whom this information was documented)	9/27	19/39	0.322
Co-morbidity associated with immunosuppression/total number of patients for whom this information was documented	6/27	16/51	0.555
Known to be HIV-positive/number of patients for whom test results were documented	0/18	0/23	Not applicable
Microbiological confirmation (number of patients with microbiological confirmation/number of patients for whom this information was documented)	18/24	45/46	0.005
Immunodiagnostic test result of positive/number of patients tested	24/27	35/52	0.030

**Table 2 tropicalmed-10-00070-t002:** Personal characteristics of patients with isolated lymph node tuberculosis and patients with miliary and/or central nervous system tuberculosis from less-than-high-income countries.

	Patients with Lymph Node Tuberculosis	Patients with Miliary and/or Central Nervous System Tuberculosis	*p*-Value for Comparison Between Groups
Age (years, median (range))	20.0 (0.5–52.0)	14.5 (0.04–69)	0.556
Gender (male/total number of patients for whom this information was documented)	15/25	13/29	0.401
Co-morbidity associated with immunosuppression/total number of patients for whom this information was documented	2/22	3/38	1.000
Known to be HIV-positive/number of patients for whom test results were documented	2/12	1/25	0.240
Microbiological confirmation(number of patients with microbiological confirmation/number of patients for whom this information was documented)	15/22	35/42	0.282
Immunodiagnostic test result of positive/number of patients tested	18/25	46/53	0.203

**Table 3 tropicalmed-10-00070-t003:** Personal characteristics and laboratory parameters in patients with positive versus negative immunodiagnostic test results.

	Patients with Positive Immunodiagnostic Tests (n = 120)	Patients with Negative Immunodiagnostic Tests (n = 37)	*p*-Value
Age (years, median (range))	30 (0.04–89)	17 (0.41 to 87)	0.049
Gender (male/patients for whom this information was available)	50/110	19/37	0.666
Co-morbidity associated with immunosuppression/total number of patients for whom this information was documented	27/105	9/24	0.363
Microbiological confirmation (number of patients with microbiological confirmation/number of patients for whom this information was documented)	102/114	23/32	0.026
Known to be HIV-positive/number of patients for whom test results were documented	2/53	1/21	1.000
Patients from high-income countries/number of patients for whom this information was documented	58/120	18/37	0.877
Hemoglobin level (g/dL, mean (standard deviation))	11.1 (2.7) ^1^	11.0 (1.8) ^1^	0.929
Lymphocyte count (per microliter, median (range))	1985 (800 to 5100) ^2^	2400 (169 to 24,000) ^2^	0.730
CRP level (mg/dL, median (range))	5.7 (0–51) ^3^	11.3 (1–226) ^3^	0.050
ESR (mm/h, median (range))	42.5 (5 to 120) ^4^	32 (6 to 109) ^4^	0.767

^1^ A hemoglobin level was available for 12 patients with positive and 15 patients with negative immunodiagnostic test results. ^2^ A lymphocyte count was available for 20 patients with positive and 15 patients with negative immunodiagnostic test results. ^3^ A CRP level was available for 32 patients with positive and 15 patients with negative immunodiagnostic test results. ^4^ An ESR level was available for 18 patients with positive and 13 patients with negative immunodiagnostic test results.

## Data Availability

The data of this study are available on request and included in most of the quoted publications.

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
