# Peer review of "Sensitivity of Immunodiagnostic Tests in Localized Versus Disseminated Tuberculosis—A Systematic Review of Individual Patient Data"

_tropicalmed, 2025, doi:10.3390/tropicalmed10030070_

Round 1

Reviewer 1 Report

Comments and Suggestions for Authors

This study presents interesting and highly relevant insights for clinical practice, with results that could significantly impact both scientific knowledge and clinical application. However, a correction in methodology is necessary.

The main objective of this study is to investigate the sensitivity and specificity of immunodiagnostic tests for confirming active tuberculosis in three clinical situations: tuberculous lymphadenitis, tuberculous meningitis, and pulmonary miliary tuberculosis. Additionally, the study will differentiate between children and adults, as their diagnostic approaches in clinical practice may vary.

To effectively address these objectives, it is essential to include only confirmed cases of active tuberculosis, defined as follows: (1) positive culture for Mycobacterium tuberculosis, (2) detection of acid-fast bacilli along with a positive polymerase chain reaction (PCR) for M. tuberculosis, or (3) histopathological findings characterized by epithelioid cell granulomas along with positive PCR for M. tuberculosis.

By examining these specific cases, the study aims to clarify the value of immunodiagnostic tests in diagnosing active tuberculosis. Furthermore, analyzing the data separately for adults and children will enhance the applicability of this information in clinical practice. Ultimately, this is the primary goal of any scientific research.

Author Response

Reviewer 1:

Comment 1: This study presents interesting and highly relevant insights for clinical practice, with results that could significantly impact both scientific knowledge and clinical application. However, a correction in methodology is necessary.The main objective of this study is to investigate the sensitivity and specificity of immunodiagnostic tests for confirming active tuberculosis in three clinical situations: tuberculous lymphadenitis, tuberculous meningitis, and pulmonary miliary tuberculosis. Additionally, the study will differentiate between children and adults, as their diagnostic approaches in clinical practice may vary.To effectively address these objectives, it is essential to include only confirmed cases of active tuberculosis, defined as follows: (1) positive culture for Mycobacterium tuberculosis, (2) detection of acid-fast bacilli along with a positive polymerase chain reaction (PCR) for M. tuberculosis, or (3) histopathological findings characterized by epithelioid cell granulomas along with positive PCR for M. tuberculosis.

Our reply to comment 1: The reviewer suggested to include only microbiologically (culture or PCR confirmed) confirmed cases of active tuberculosis:To include only microbiologically confirmed cases of tuberculosis does not conform with internationallly agreed case definitions of active tuberculosis. A significant proportion of cases internationally accepted as active tuberculosis are not microbiologically confirmed particularly in children. Only including microbiologically confirmed cases would undermine the validity of our study and make it useless for clinical practice and scientific research, which depends on reproducibility of the research by adherence to internationally accepted case definitions. It also would render our important finding of a likely influence of bacterial load on IGRA positivity impossible to detect.We have therefore not changed our methodology.

Comment 2: By examining these specific cases, the study aims to clarify the value of immunodiagnostic tests in diagnosing active tuberculosis. Furthermore, analyzing the data separately for adults and children will enhance the applicability of this information in clinical practice. Ultimately, this is the primary goal of any scientific research.

Our reply to comment 2: We agree and have now presented a comparison of sensitivity of immunodiagnostic testing in localized versus disseminated tuberculosis in children and adults separately. We have change methods, results and discussion sections accordingly.

Reviewer 2 Report

Comments and Suggestions for Authors

Eisenhut et al. present a very interesting systematic review of IGRA positivity in TB lymphadenitis vs. disseminated TB. I think the findings would be of value to both clinicians and laboratorians involved with TB diagnosis and care. 

I think the paper is very well done and don't have any major suggestions for improvement. 

Some minor suggestions:

  1. Introduction - the statement that immunodiagnostic testing IS a useful adjunct in diagnosis (line 42) sounds too strong to me for both the evidence cited, as well as my background info. Would recommend to edit to "immunodiagnostic testing MAY BE a useful adjunct in diagnosis of active TB..." In my personal opinion over-recommending/over-praising immunodiagnostic testing sometimes has the effect of reducing attempts for microbiological diagnosis.
  2. For "definition of tuberculous meningitis", the first supporting criterion lists a scenario where CSF culture is sterile, but there is no mention of how positive cultures and/or positive molecular tests were going to be included in patient selection - please update this section for these 
  3. For the Exclusion criteria, the 2nd one is test result obtained prior to illness onset - is it at any point prior to illness onset (e.g. even 1 day prior), vs. more than some particular period of time prior to illness onset? 
    How was "illness onset" defined? - prior to symptoms onset, prior to diagnosis?
  4. For Appendix Table 3, can you include a comparison of whether immunosuppressed patients were represented at a significantly higher percentage in the cohort form high income countries vs. not high income countries. 

Author Response

Reviewer 2:

Eisenhut et al. present a very interesting systematic review of IGRA positivity in TB lymphadenitis vs. disseminated TB. I think the findings would be of value to both clinicians and laboratorians involved with TB diagnosis and care. 

I think the paper is very well done and don't have any major suggestions for improvement. 

Some minor suggestions:

  1. Introduction - the statement that immunodiagnostic testing IS a useful adjunct in diagnosis (line 42) sounds too strong to me for both the evidence cited, as well as my background info. Would recommend to edit to "immunodiagnostic testing MAY BE a useful adjunct in diagnosis of active TB..." In my personal opinion over-recommending/over-praising immunodiagnostic testing sometimes has the effect of reducing attempts for microbiological diagnosis.

Our reply to comment 1: We have changed this sentence following the reviewer comments.

  1. For "definition of tuberculous meningitis", the first supporting criterion lists a scenario where CSF culture is sterile, but there is no mention of how positive cultures and/or positive molecular tests were going to be included in patient selection - please update this section for these 

Our reply to comment 2: We have clarified that we included both patients with definite and highly probable tuberculous meningitis. The latter does not require positive cultures and/or positive molecular tests and relies on a combination of supporting criteria.

  1. For the Exclusion criteria, the 2nd one is test result obtained prior to illness onset - is it at any point prior to illness onset (e.g. even 1 day prior), vs. more than some particular period of time prior to illness onset? How was "illness onset" defined? - prior to symptoms onset, prior to diagnosis?

Our reply to comment 3: It is at any point prior to illness onset which was defined as symptom onset. We have mentioned this now in the text. 

  1. For Appendix Table 3, can you include a comparison of whether immunosuppressed patients were represented at a significantly higher percentage in the cohort form high income countries vs. not high income countries.

Our reply to comment 4: We have now included a comparison of percentage of immunosuppressed patients between high versus less than high income countries in the result section.

Reviewer 3 Report

Comments and Suggestions for Authors

Estimated Authors,

I've read with great interest your paper entitled "Sensitivity of immunodiagnostic tests in localized versus disseminated tuberculosis- a systematic review of individual patient data". In this paper, Authors have reported data from a total of 157 patients (52 patients with Lymph node tuberculosis; 105 with miliary tuberculosis and/or tuberculous meningitis). Individual cases were retrieved and summarized, suggesting that a microbiological confirmation of the TB infection is more common among patients with IGRA+ test. 

Unfortunately, the present paper cannot be accepted for publication in its current stage of development because of severe shortcomings which I will briefly report in the following lines.

1) DESIGN: the present paper has been apparently designed according to PRISMA statement, but Authors have not provided several important items, including: (a) the searching strategy: by lacking search strategy, the potential replicability of results is questionable, impairing the very same aims of a systematic review; (b) the risk of bias analysis is quite confusingly reported: statements such as "We think there was positive reporting bias for the case reports included in this review" and "There is no reason to believe that this differed between the groups with localized versus disseminated tuberculosis" are not sustantiated by actual data, being therefore questionable, both in their reporting and in their content. Authors are welcome in performing an accurate ROB analysis through some options that have been made available by previous researches (e.g. Murad, M.H.; Sultan, S.; Haffar, S.; Bazerbachi, F. Methodological Quality and Synthesis of Case Series and Case Reports. Evid. Based Med. 2018, 23, 60–63). 

2) OBJECTIVES: according to the rationale, the study was designed in order to "perform a systematic review of individual patient data comparing immunodiagnostic test sensitivity in patients with localized versus disseminated tuberculosis in patients from high income countries", to "perform a systematic review of individual patient data comparing immunodiagnostic test sensitivity in patients with localized versus disseminated tuberculosis in patients from less than high income countries", to "identify risk factors for false negative immunodiagnostic tests in patients with active tuberculosis".

To begin with, there is a clear dichotomy between the main title (Authors did not perform any analysis of diagnostic performances, including sensitivity) and the pooled results. In fact, Authors have reported only the following three outcomes:

a) occurrence of personal characteristics of patients with isolated lymphnode tuberculosis and patients with miliary and/or central nervous system tuberculosis (Table 1+2)

b) Personal characteristics and laboratory parameters in patients with positive versus with negative immunodiagnostic testing (Table 3)

Please revise in order to provide a consistent reporting.

3) SUMMARY OF MAIN DATA: being individual data not provided, it is also unclear how main data were actually identified and pooled. According to the main text, "we extracted data on age, gender, country of origin, IGRA for M. tuberculosis infection and HIV test result at diagnosis, results of imaging, histopathology and results of nucleic acid amplification tests (NAAT) and culture of any site obtained. Microbiological confirmation of tuberculosis was defined as positive NAAT or culture result. Any laboratory parameters obtained were recorded. The response to treatment was recorded". In table 1, 2, and 3 such a data are only partially and incompletely reported and discussed. Please either improve data reporting or provide a more accurate description within methods section.

Comments on the Quality of English Language

Although the quality of the English text is sufficient, the flow of the main text may be improved by an extensive revision.

Author Response

Reviewer 3

Comment 1) DESIGN: the present paper has been apparently designed according to PRISMA statement, but Authors have not provided several important items, including: (a) the searching strategy: by lacking search strategy, the potential replicability of results is questionable, impairing the very same aims of a systematic review.

Our reply to comment 1: For the searching strategy we refer the reviewer to page 4, line 149 to line 154 entitled: “Identifying studies-information sources and search strategy:”

Comment 2) (b) the risk of bias analysis is quite confusingly reported: statements such as "We think there was positive reporting bias for the case reports included in this review" and "There is no reason to believe that this differed between the groups with localized versus disseminated tuberculosis" are not sustantiated by actual data, being therefore questionable, both in their reporting and in their content. Authors are welcome in performing an accurate ROB analysis through some options that have been made available by previous researches (e.g. Murad, M.H.; Sultan, S.; Haffar, S.; Bazerbachi, F. Methodological Quality and Synthesis of Case Series and Case Reports. Evid. Based Med. 2018, 23, 60–63). 

Our reply to comment 2: We have now deleted the sentences referred to by the reviewer. The important items contained in the paper by Murad et al. like selection, causality, ascertainment and reporting were implicitly covered in our inclusion and exclusion criteria and are therefore not useful to apply again in a post selection analysis.

Comment 3)OBJECTIVES: according to the rationale, the study was designed in order to "perform a systematic review of individual patient data comparing immunodiagnostic test sensitivity in patients with localized versus disseminated tuberculosis in patients from high income countries", to "perform a systematic review of individual patient data comparing immunodiagnostic test sensitivity in patients with localized versus disseminated tuberculosis in patients from less than high income countries", to "identify risk factors for false negative immunodiagnostic tests in patients with active tuberculosis".To begin with, there is a clear dichotomy between the main title (Authors did not perform any analysis of diagnostic performances, including sensitivity) and the pooled results. In fact, Authors have reported only the following three outcomes:

  1. a) occurrence of personal characteristics of patients with isolated lymphnode tuberculosis and patients with miliary and/or central nervous system tuberculosis (Table 1+2)
  2. b) Personal characteristics and laboratory parameters in patients with positive versus with negative immunodiagnostic testing (Table 3)

Please revise in order to provide a consistent reporting.

Our reply to comment 3: The main title of the manuscript is “Sensitivity of immunodiagnostic tests in localized versus disseminated tuberculosis- a systematic review of individual patient data”We reported the data on sensitivity in the last rows of table 1 and 2 and in form of percentage positive results in the text of results and abstract. We have now added the word sensitivity to make the connection with the title more obvious.

Comment 4)SUMMARY OF MAIN DATA: being individual data not provided, it is also unclear how main data were actually identified and pooled. According to the main text, "we extracted data on age, gender, country of origin, IGRA for M. tuberculosis infection and HIV test result at diagnosis, results of imaging, histopathology and results of nucleic acid amplification tests (NAAT) and culture of any site obtained. Microbiological confirmation of tuberculosis was defined as positive NAAT or culture result. Any laboratory parameters obtained were recorded. The response to treatment was recorded". In table 1, 2, and 3 such a data are only partially and incompletely reported and discussed. Please either improve data reporting or provide a more accurate description within methods section

Our reply to comment 4: We have now elaborated (see page 4 under the heading data collection process) that researchers retrieved the data from the manuscripts identified and transferred them onto excel files and that co-researchers contacted for provision of individual patient data were also asked to provide excel files with anonymized data specified in the pre-registered protocol of the study. Table 1, 2 and 3 don’t contain all the data because some of them were only required to confirm eligibility and not relevant for comparison of groups.

Round 2

Reviewer 1 Report

Comments and Suggestions for Authors

Unfortunately, the authors have ignored my most important comment:

„To effectively address these objectives, it is essential to include only confirmed cases of active tuberculosis, defined as follows: (1) positive culture for Mycobacterium tuberculosis, (2) detection of acid-fast bacilli along with a positive polymerase chain reaction (PCR) for M. tuberculosis, or (3) histopathological findings characterized by epithelioid cell granulomas along with positive PCR for M. tuberculosis.“

Unless the methodology is corrected, the paper cannot be published because it will be misleading.

Author Response

Reviewer 1

Comment 1

Unfortunately, the authors have ignored my most important comment:

„To effectively address these objectives, it is essential to include only confirmed cases of active tuberculosis, defined as follows: (1) positive culture for Mycobacterium tuberculosis, (2) detection of acid-fast bacilli along with a positive polymerase chain reaction (PCR) for M. tuberculosis, or (3) histopathological findings characterized by epithelioid cell granulomas along with positive PCR for M. tuberculosis.“

Unless the methodology is corrected, the paper cannot be published because it will be misleading.

Our reply to comment 1 (as in our previous reply to this comment):

The reviewer suggested to include only microbiologically (culture or PCR confirmed) confirmed cases of active tuberculosis:To include only microbiologically confirmed cases of tuberculosis does not conform with internationallly agreed case definitions of active tuberculosis. A significant proportion of cases internationally accepted as active tuberculosis are not microbiologically confirmed particularly in children. Only including microbiologically confirmed cases would undermine the validity of our study and make it useless for clinical practice and scientific research, which depends on reproducibility of the research by adherence to internationally accepted case definitions. It also would render our important finding of a likely influence of bacterial load on IGRA positivity impossible to detect.We have therefore not changed our methodology.

Reviewer 3 Report

Comments and Suggestions for Authors

Estimated Authors,

I've appreciated the efforts to cope with my previous comments.

Only a minor suggestion before the eventual acceptance, that is please provide the 95% confidence intervals of sensitivity estimates for guaranteeing its more appropriate appraisal.

Author Response

Comment 1

Estimated Authors,

I've appreciated the efforts to cope with my previous comments.

Only a minor suggestion before the eventual acceptance, that is please provide the 95% confidence intervals of sensitivity estimates for guaranteeing its more appropriate appraisal.

 Our reply to comment 1

We have in response to the reviewer suggestion included 95% confidence intervals for the sensitivity estimates in our result section.